

# Exploring ethical monitoring of physical activity behaviors among adults: a Smart Platform study operationalizing digital citizen science

Caitlin Hung[1], Sheriff Tolulope Ibrahim[1,2] and Tarun Reddy Katapally[1,2,3]

[1] DEPtH Lab, School of Health Studies, Faculty of Health Sciences, University of Western Ontario, London, Ontario, Canada
[2] Children's Health Research Institute, Lawson Health Research Institute, London, Ontario, Canada
[3] Department of Epidemiology and Biostatistics, Schulich School of Medicine and Dentistry, University of Western Ontario, London, Ontario, Canada

Corresponding author
Tarun Reddy Katapally,
tarun.katapally@uwo.ca

## ABSTRACT

**Background**. According to the World Health Organization, 27% of adults do not meet the recommended daily levels of physical activity (PA), making accurate PA measurement essential for informing evidence-based policies. This study explores ethical engagement with citizens through their ubiquitous digital tools (*i.e.*, smartphones) to examine variations between retrospectively and prospectively reported PA behaviors within the same cohort.

**Methods**. This study is part of the Smart Platform, a digital citizen science initiative for ethical monitoring and real-time intervention. Data were collected from 118 adults who participated over eight consecutive days, including both weekdays and weekends. Prospective PA was assessed using time-triggered ecological assessments, while retrospective PA was measured using a modified, time-triggered, smartphone-based validated tool. Paired sample $t$-tests were used to compare retrospective and prospective PA. Linear regression models examined associations between socio-demographic and contextual factors and both types of PA reporting. Analyses were conducted for the overall sample and by gender (male *vs.* female).

**Results**. Participants consistently reported higher PA through retrospective measures compared to prospective ones ($p < 0.001$). In the overall sample, one significant association was found in the retrospective model, while three were identified in the prospective model. Among males, those who engaged in PA for fun or to maintain physical health reported higher retrospective PA, though this was not significant in the prospective model. In contrast, female participants who engaged in PA for fun reported higher PA in both retrospective and prospective models.

**Conclusions**. Although exploratory, early findings suggest that repeated, prospective assessments *via* ubiquitous digital devices may enhance the validity and reliability of PA measurement. As citizen-owned digital tools become increasingly widespread, ethically leveraging big data through digital citizen science offers a promising approach to improve PA monitoring and support public health efforts.

## INTRODUCTION

Non-communicable diseases (NCDs), including obesity, hypertension, type 2 diabetes, and cancer, are one of the leading causes of death worldwide (*World Health Organization, 2023*). Physical activity (PA) has been proven to prevent and manage NCDs (*WHO, 2022*), yet a significant number of men (25%) and women (33%) worldwide do not meet the recommended levels of PA (*WHO, 2022*). These individuals have a 20% to 30% higher risk of mortality in comparison to those who meet the PA guidelines (*WHO, 2022*; *Pryor, Silva & Melchior, 2017*). Given the significant role of PA in mitigating NCDs and improving mental health, it is imperative to accurately measure PA to understand behavior patterns.

Traditional retrospective surveys often rely on individuals' recall ability, which may lead to overestimation (*Dowd et al., 2018*; *Silsbury, Goldsmith & Rushton, 2015*). Measurements of PA in adults obtained through ecological prospective assessments (EPAs)—which involve real-time, continuous data collection using smartphones—can be more accurate and reliable than those obtained through retrospective traditional surveys (*Katapally & Chu, 2020*). The digital era provides innovative opportunities to advance accurate measurement of PA behaviors by re-purposing ubiquitous tools such as smartphones (*Laranjo et al., 2021*). However, thus far, non-digital retrospective measures have primarily been employed in PA behavioral research which is prone to recall bias (*Moreno-Serra et al., 2022*; *Zang et al., 2023*).

Recall bias occurs due to inaccurate recall in reporting of past behaviors, often leading to over- or under-estimation of PA levels (*Althubaiti, 2016*). Recalling various PA behaviors over time can be challenging for individuals, resulting in significant recall bias, especially comparing longer to shorter recall periods (*Althubaiti, 2016*; *Prince et al., 2008*). For instance, in a study comparing self-reported and objectively measured PA in undergraduate students, self-reported vigorous and moderate-to-vigorous PA were significantly higher, while sitting time and moderate PA were significantly lower in objectively measured data (*Nelson, Katie & Vella, 2024*). The use of objective measures, including accelerometers, gyroscopes, pedometers, global positioning systems, and other movement sensors, can reduce recall biases. However, objective measures can be expensive (*Aparicio-Ugarriza et al., 2015*), time-consuming (*Skender et al., 2016*) and challenging to implement across diverse populations (*Lonsdale et al., 2016*). Moreover, objective measures often overlook environmental interactions (*Zhang, Zhou & Kwan, 2019*) and social context (*Joffer et al., 2019*) that can be captured through repeated measurement of participant perceptions.

EPAs deployed *via* citizen-owned digital devices, can capture changes in health behaviors and outcomes over time and address some of the limitations associated with current approaches to PA measurement (*Mok et al., 2019*). For instance, EPAs can enable daily participant reporting of PA *via* their smartphones (*DeVries, Baselmans & Bartels, 2021*). Moreover, EPAs can also capture critical social and physical contexts of PA (*Katapally & Chu, 2020*)—data that are important to inform behavioral interventions (*Perski et al., 2022*). A previous study conducted in a multi-ethnic Asian cohort using EPAs found that both park use, and leisure-time PA were independently associated with reduced evening stress (*Park et al., 2022*). Additionally, EPAs have been utilized to examine home

environmental factors that influence PA-related behaviors, such as in a study of mothers of toddlers from low-income households, which found that PA levels were higher when toddlers were outdoors, surrounded by other children, or actively interacting with their mothers compared to other settings (*Campbell et al., 2021*). Nevertheless, although EPAs offer a convenient and contextually rich approach to capturing PA, data privacy and security concerns may arise from collecting prospective data through citizen-owned smartphones (*Ozair et al., 2015*).

Given the significant role of PA in mitigating NCDs, accurate measurement is essential for understanding behavior patterns (*Paleco et al., 2021*). However, traditional self-report methods often suffer from recall bias and lack real-time context. To address these limitations, this study adopts an EPA and digital citizen science approach, which leverages smartphone-based data collection and feedback mechanisms to capture PA activities in real-time and natural environments (*Perski et al., 2022*). This method enhances reporting accuracy, where participants contribute to all aspects of the research *via* their ubiquitous digital devices. It offers additional advantages by enabling large-scale, inclusive participation regardless of geographic or socioeconomic constraints (*Paleco et al., 2021*). It also promotes ethical and transparent data practices by allowing participants to co-design and co-implement data collection processes (*Ozair et al., 2015*; *Katapally & Ibrahim, 2023*), maintain control over their data, and request data deletion at any time (*Katapally et al., 2018*). This approach embodies the concept of ethical surveillance, which differs from conventional surveillance by prioritizing participant autonomy, informed consent, and transparency (*WHO, 2024*). Rather than passive monitoring, ethical surveillance actively involves participants in decision-making about their data, fostering trust and accountability in digital data collection practices (*Buchbinder et al., 2020*). This participatory approach offers a more precise, scalable, and contextually relevant alternative to traditional PA monitoring. Additionally, PA engagement is shaped by motivational, environmental, and socio-demographic factors (*Corder, Ogilvie & Van Sluijs, 2009*), and occurs at varying intensities (*Luong et al., 2021*), each presenting unique measurement challenges. These complexities reinforce the need for innovative, participant-driven approaches that leverage digital tools to enhance reporting accuracy and engagement (*Perski et al., 2022*; *Corder, Ogilvie & Van Sluijs, 2009*; *Luong et al., 2021*; *Lee & Paffenbarger, 2000*; *Jakicic et al., 1997*; *Pharr, Lough & Terencio, 2020*).

While previous studies have compared retrospective and prospective PA measurements in sub-populations (*Ibrahim, Hammami & Katapally, 2023*; *Andorko et al., 2019*), few studies have findings that apply to the general adult population. Additionally, many studies have primarily focused on one type of PA measurement, such as objective data (*Silfee et al., 2018*; *Evenson, Buchner & Morland, 2012*), retrospective recall (*Kuutti et al., 2023*; *Schmidt et al., 2006*), or ecological momentary assessment (*Campbell et al., 2021*), without comparing these approaches directly, limiting our understanding of the differences in these methods of PA measurement. To address these gaps, this study aims to compare two measures of PA, retrospective and prospective, in an adult population.
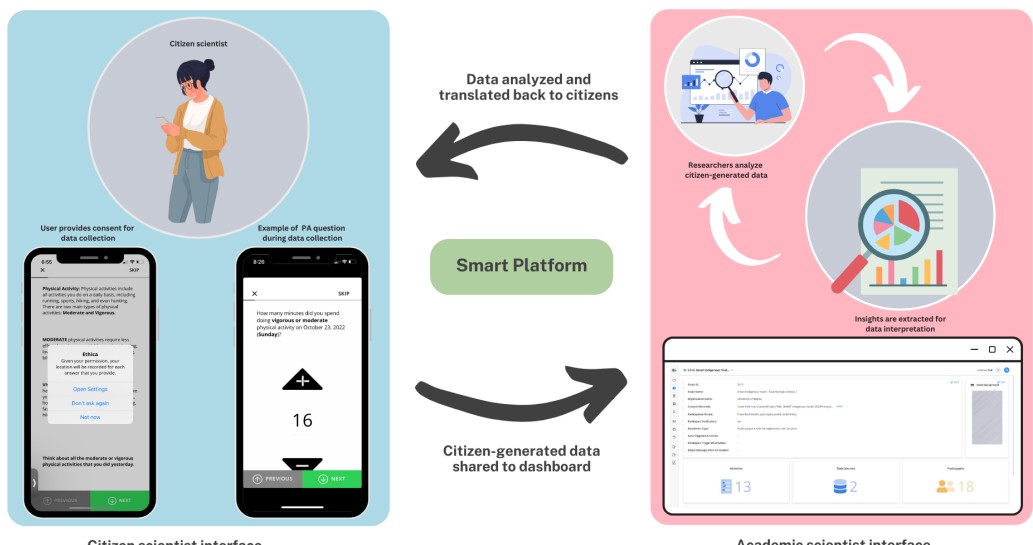

**Figure 1  The Smart Platform showing the citizen scientist and academic scientist interfaces.**

## METHODS

### Study design

This study is part of the Smart Platform (Fig. 1), a digital epidemiological initiative (*Katapally, 2019*) that connects a custom-built smartphone application (app) downloaded by citizen scientists on their personal smartphones with a backend dashboard utilized by academic scientists. The Smart Platform integrates two primary interfaces for human–computer interaction: the citizen interface (smartphone app) and the researcher interface (digital dashboard), which displays aggregated and anonymized big data relayed from citizens. The Smart Platform supports real-time, remote human-to-human interaction, enabling anonymized communication, which can be initiated by either citizen scientists or academic scientists. This study's design combines cross-sectional time-triggered validated survey measures with longitudinal time-triggered EPAs to capture participants' PA behavior over eight consecutive days, including weekdays and weekends, *via* the smartphone app. The time-triggering mechanism deployed the EPA notifications at 8 PM each day, with responses allowed until midnight to enable participants to report daily behaviors. These notifications were single-triggered, meaning they were sent once per day without retriggering (*Katapally et al., 2018*). Ethics approval for this study was granted by the Research Ethics Boards of the Universities of Saskatchewan and Regina (REB #2017-029).

### Recruitment process

This study included adults located in Regina, Saskatchewan, Canada. Participants were recruited through a combination of online (social media) and in-person recruitment sessions (*Katapally, Hammami & Chu, 2021*). The in-person recruitment sessions were carried out at the Universities of Regina and Saskatchewan, as well as community centers located in different neighborhoods in each city. During recruitment sessions, the research
team explained the study to participants and answered questions from study participants. The research team provided step-by-step guidance on how to download, install, and use the app to support individuals with varying levels of digital literacy (*Katapally et al., 2018*). Interested participants downloaded the custom-built smartphone app onto their smartphones, provided informed consent *via* the app, (≥18 years) before joining the study. In this study, participants were "citizen scientists" due to their collaboration with the researchers throughout the study (*Katapally, Hammami & Chu, 2021*). Citizen scientists engaged with researchers in real-time through the app and also had access to the research team *via* email, allowing for continuous and flexible two-way communication. For example, participant feedback on the timing of the time-triggered EPAs led researchers to extend the expiration window of EPAs, thereby enhancing flexibility and improving data reporting. Although the feedback was not formally recorded, it directly informed procedural refinements, demonstrating the study's participant-centered and adaptive approach. Extending the EPA's expiration period improved flexibility and data quality without compromising compliance, as participants consistently responded within the designated timeframe. These changes also strengthened the ethical transparency and contextual relevance of the data collection process.

## Measures

Using the custom-built app, citizen scientists provided data over eight consecutive days including weekdays and weekends, which included sociodemographic variables as well as time-triggered cross-sectional measures that captured retrospective PA and time-triggered EPAs that captured prospective PA (Fig. 2). PA measures were ascertained using the International Physical Activity Questionnaire (IPAQ), which was adapted to be deployed *via* smartphones. IPAQ has demonstrated a strong test-retest reliability, with $p = 0.81$ (95% CI [0.79–0.82]) for the long form, supporting its consistency in measuring PA across diverse populations. The IPAQ is a suitable tool for monitoring PA levels among adults aged 18 to 65 across various settings, making it an effective instrument for large-scale PA monitoring and research (*Craig et al., 2003*). Participants' responses that included "strongly agree", "agree", "strongly disagree", "disagree", "neutral" and "does not apply", were dichotomized into "agree" and "disagree". The agree category included "strongly agree", and "agree" while the disagree category included "strongly disagree", "disagree", "neutral" and "does not apply". The low response count in the "neutral" and "does not apply" categories warranted its categorization to ensure sufficient sample sizes for meaningful analysis (*Van Dusen & Nissen, 2020*), and analytical robustness (*Shibata et al., 2009*; *Yıldızer et al., 2018*; *Ball et al., 2010*; *Wen, Kandula & Lauderdale, 2007*). For gender, only male and female categories were included in the analysis, as no participants selected "transgender," "other (please specify)," or "prefer not to disclose." Employment status was categorized as employed full-time and not employed full-time, as previous studies have identified associations between full-time employment and PA levels (*Lindsey et al., 2021*; *Ryde et al., 2020*; *Li et al., 2024*).

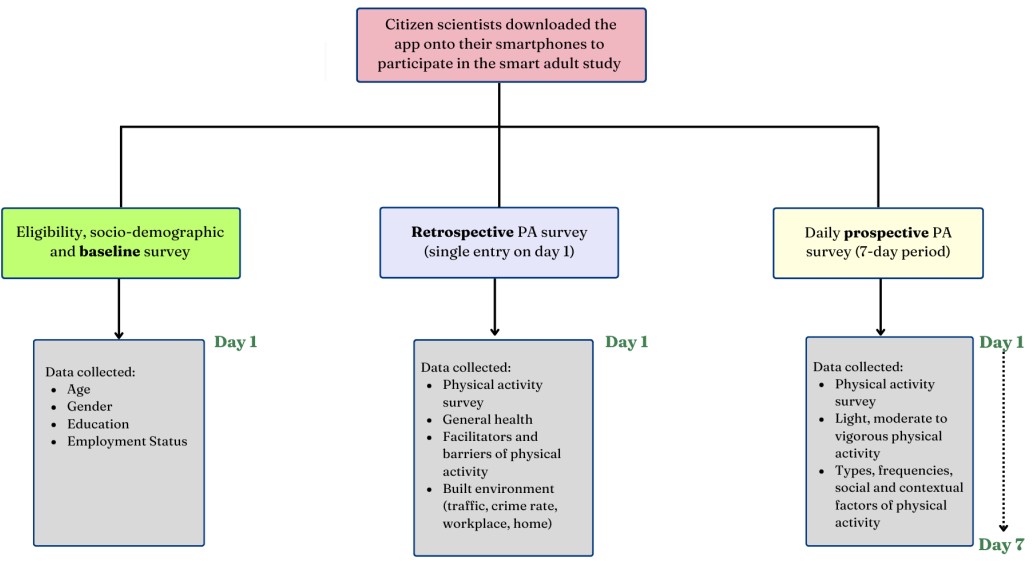

**Figure 2** Flow diagram representing process of study deployment across various measures.

### Retrospective PA (dependent variable)

Citizen scientists reported retrospective PA through a modified version of the IPAQ (*Craig et al., 2003*). The modification allowed for digital deployment of the survey and reporting retrospective PA data in four domains (work, transportation, home, and recreation), where citizen scientists were asked to report PA that was at least 10 minutes or more over the past eight days. From this, the mean retrospective PA (minutes/day) was derived.

### Prospective PA (dependent variable)

Adult citizen scientists' prospective PA information was obtained *via* daily time-triggered EPAs throughout the study period. These EPAs were designed to use skip pattern questions to capture PA accumulation prospectively. After defining what constitutes PA, EPAs were deployed to ask the following questions: (1) What type of physical activity did you do today? (Multiple choice); (2) How many minutes did you spend doing this physical activity? (Open-ended, skip pattern). From these questions, we derived the mean prospective PA (minutes/day).

### Socio-demographic factors

Gender identity was determined by the question, "What is your gender?" with five responses including: "male," "female," "transgender," "other (please specify)," and "prefer not to disclose." The educational level of the participants was assessed by having them indicate the highest level of education by offering four choices: "some secondary/high school," "completed secondary/high school," "some post-secondary (university or college)," and "completed post-secondary (university or college)". These responses were grouped into two educational levels: (1) "some secondary/high school" and "completed secondary/high school" were labeled as "at least secondary school" (2) "some post-secondary (university or college)" and "completed post-secondary (university or college)" were labeled as

"university and above". Employment status was determined by selecting employed full-time, employed full-time (self-employed), employed part-time, employed part-time (self-employed), unemployed, receiving social assistance, receiving disability or retirement pension, student, and other (*e.g.*, seasonal temporary). The options were further categorized into employed full-time and not employed full-time.

### Health motivation for PA (independent variables)

Participants were asked to answer the following question to assess the impact of health motivation on PA: "I undertake physical activity to improve cardiovascular fitness" and "I undertake physical activity to maintain physical health". Participants were given the response options of "strongly agree", "agree", "neutral", "disagree", "strongly disagree" or "does not apply". After data collection, responses were dichotomized into "agree" (including strongly agree and agree) and "disagree" (including neutral, disagree, strongly disagree, and does not apply). These variables were derived from the Motives for Physical Activities Measure—Revised, a validated scale designed to assess the motivational factors underlying PA (*Richard et al., 2025*).

### Recreational motivation for PA (independent variable)

Recreational motivation for PA was captured by asking the following question: "I undertake physical activity because it's fun". Participants were given the response options of "strongly agree", "agree", "neutral", "disagree", "strongly disagree" or "does not apply". After data collection, responses were dichotomized into "agree" (including strongly agree and agree) and "disagree" (includes neutral, disagree, strongly disagree, and does not apply). These variables were derived from the Motives for Physical Activities Measure—Revised, a validated scale designed to assess the motivational factors underlying PA.

### Environmental factors (independent variables)

The influence of home facilities on PA was captured through the following statements: "I have facilities/equipment to exercise in my home." and "I have space to exercise at home." Participants were given the response options of "strongly agree", "agree", "neutral", "disagree", "strongly disagree" or "does not apply". After data collection, responses were dichotomized into "agree" (including strongly agree and agree) and "disagree" (including neutral, disagree, strongly disagree, and does not apply). These variables were derived from the Physical Activity and Media Inventory, a validated instrument designed to assess the availability, type, and accessibility of PA-related equipment and media in the home environment (*Sirard et al., 2008*).

### Multi-level inclusion and exclusion criteria with bias mitigation strategies

The inclusion criteria to determine the final sample were dependent on participants completing the retrospective PA measures and providing prospective EPAs on at least one day. To ensure methodological rigor and address the issue of outliers, participants who reported PA of less than 10 (minutes/day) (*Ryde et al., 2020*) or more than three standard deviations above the mean were excluded from the analyses in this study (Fig.

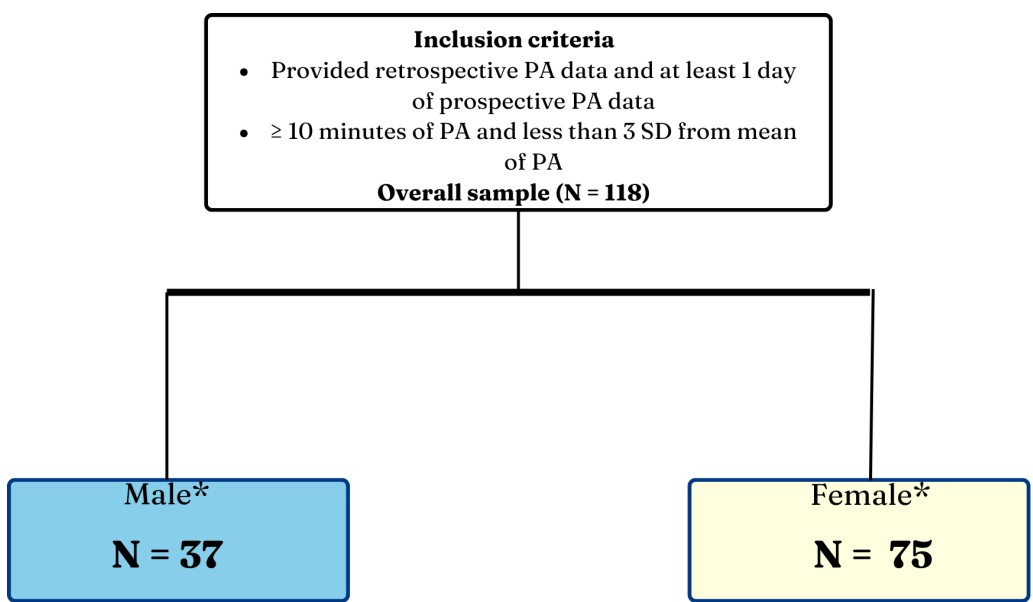

**Figure 3 Inclusion criteria of participants.** *The sum of male and female do not add up to the overall sample as some participants did not provide gender data.

3). The exclusion of participants with extreme PA values follows IPAQ guidelines (*Ryde et al., 2020*), as such values are likely due to measurement errors, or unrealistic self-reports, rather than true activity levels. This approach reduces statistical bias, preserves model assumptions, and enhances the interpretability of associations (*Li et al., 2024*). Since the surveys were deployed *via* smartphones, potential biases such as social desirability, where participants may report socially desirable behaviors or underreport undesirable ones (*Teh et al., 2023*) were mitigated by allowing participants to respond privately without the interviewer's influence. This digital approach reduced pressure to provide socially desirable answers, enhancing response accuracy. Additionally, the use of time-triggered EPAs enabled real-time PA tracking, minimizing reliance on retrospective recall and improving data reliability.

## Data risk management

The custom-built app used in this study was designed to prioritize confidentiality, data privacy, and security through a multi-layered protection framework. All participant data were encrypted before transmission to a secure cloud server, ensuring robust protection against unauthorized access. To prevent breaches, encryption protocols were implemented to guard against risks such as eavesdropping, man-in-the-middle attacks, and rogue hotspots. The app's "snooze" feature empowered participants to halt data transmission at any time, ensuring complete control over their information sharing. No personally identifiable data, such as website visits, GPS location, or contact lists, were accessed or stored. To further safeguard privacy, all Media Access Control addresses were irreversibly anonymized *via* a hashing algorithm. Participants received comprehensive explanations of these protections both during in-person recruitment and through the

app itself. Additionally, before any encrypted data capture or storage occurred, each participant provided informed consent directly through the Smart Platform interface on their smartphone, confirming their agreement in advance of any data collection. Prior to any data collection or encrypted storage, informed consent was obtained *via* the Smart Platform interface on participants' smartphones, ensuring that they explicitly agreed before any data were captured (*Katapally, 2019*). Participants could revisit the consent form at any time within the app, withdraw from the study, and request permanent deletion of their data. They could also choose to upload data only when connected to Wi-Fi or when their devices were charging, minimizing potential data loss and ensuring controlled data transmission. Additionally, clear instructions on withdrawing from the study were provided within the app, and participants could contact the research team *via* dedicated Smart Platform study emails for support and inquiries. These safeguards collectively ensure that participants retain autonomy over their data throughout the study, reflecting a commitment to ethical and transparent data management, including data ownership and freedom of withdrawal from the study.

### Statistical analysis

Statistical analyses were performed using open-source statistical software analysis, R 4.2.1. Frequencies and percentages were used to describe categorical variables, while mean and standard deviation were used to describe the continuous variables used in this study. The comparison of mean retrospective and prospective measured PA was conducted through paired sample $t$-tests. Multiple linear regression models were used to ascertain the associations between retrospective and prospective measured PA with demographic and contextual factors among adults. Analyses were conducted on the overall and gender (male *vs* female) sub-samples. Sample size calculation was carried out using G*Power 3.1.9.7. The sample size calculation resulted in a sample size of 89 for a 95% power to detect a moderate effect size of 0.15 (Cohen's $f^2$). We conducted a sensitivity analysis to evaluate the impact of our exclusion criteria on study findings. All statistical significance was considered at $p < 0.05$.

## RESULTS

A total of 118 participants were included in the analyses of this study. These participants provided data on retrospective PA and at least one day of prospective PA, sociodemographic variables (age, gender, education, employment status) and independent variables (health and recreational motivations of PA, and environmental factors). The summary statistics of participants are presented in Table 1. The mean age of participants was 35.61 years old (SD = 17.25), with the majority identifying as female (66.98%) Participants predominantly received university or college degree/diploma (58.04%), 29.46% had some post-secondary education, 10.71% completed secondary/high school, while 1.79% had some secondary/high school education. Regarding employment status, 63.89% of the participants were employed but not in full-time positions, and 36.11% were employed in full-time positions.

**Table 1  Summary statistics for citizen scientists participating in this study.**

| Independent variables | Mean (SD) |
|---|---|
| Age | 35.61 (17.25) |
| **Gender** | **Percentage** |
| Male ($n = 37$) | 33.04 |
| Female ($n = 75$) | 66.96 |
| Total ($n = 112$)[a] | 100 |
| **Education** | |
| Completed secondary/high school ($n = 12$) | 10.71 |
| University or college degree/diploma ($n = 65$) | 58.04 |
| Some post-secondary (university or college) ($n = 33$) | 29.46 |
| Some secondary/high school ($n = 2$) | 1.79 |
| Total ($n = 112$)[a] | 100 |
| **Employment status** | |
| Employed full-time ($n = 69$) | 63.89 |
| Not employed full-time ($n = 39$) | 36.11 |
| Total ($n = 108$)[a] | 100 |
| **Health motivation** | |
| Improve cardiovascular fitness –Agree ($n = 105$) | 89.74 |
| Improve cardiovascular fitness –Disagree ($n = 12$) | 10.26 |
| Total ($n = 117$)[a] | 100 |
| To maintain physical health –Agree ($n = 114$) | 96.61 |
| To maintain physical health –Disagree ($n = 4$) | 3.39 |
| Total ($n = 118$) | 100 |
| **Recreational motivation** | |
| Undertake PA for fun –Agree ($n = 93$) | 79.49 |
| Undertake PA for fun –Disagree ($n = 24$) | 20.51 |
| Total ($n = 117$)[a] | 100 |
| **Environmental factors** | |
| Have facilities/equipment at home –Agree ($n = 75$) | 64.66 |
| Have facilities/equipment at home –Disagree ($n = 41$) | 35.34 |
| Total ($n = 116$)[a] | 100 |
| Space to exercise at home –Agree ($n = 79$) | 68.10 |
| Space to exercise at home –Disagree ($n = 37$) | 31.90 |
| Total ($n = 116$)[a] | 100 |

Notes.

[a]Some participants did not provide complete information; SD, Standard Deviation.

Regarding recreational motivations for undertaking PA, 79.49% of participants reported that they undertake PA for fun, whereas 20.51% reported that they do not undertake PA for fun. For health motivations, 89.74% of participants reported that they undertake PA to improve cardiovascular fitness, while 10.26% do not undertake PA to improve cardiovascular fitness. Furthermore, 96.61% of participants reported that they undertake PA to maintain their physical health, whereas 3.39% disagreed that they undertake PA to maintain their physical health. Regarding environmental factors, 64.66% reported that they have facilities/equipment to exercise in their homes, while 35.34% reported that they do

**Table 2** *t*-test results comparing mean prospective and retrospective PA in the overall and gender (male *vs.* female) sub-categories.

| | Mean | Minimum | Maximum | N | *p*-value |
|---|---|---|---|---|---|
| **Overall** | | | | | |
| **Prospective (EPA)** | 79.82 | 12.00 | 368.75 | 118 | * |
| **Retrospective (Traditional Surveys)** | 204.67 | 10.00 | 700.00 | 118 | |
| **Male** | | | | | |
| **Prospective (EPA)** | 94.73 | 12.00 | 352.00 | 37 | * |
| **Retrospective (Traditional Surveys)** | 223.60 | 25.00 | 623.00 | 37 | |
| **Female** | | | | | |
| **Prospective (EPA)** | 74.32 | 18.75 | 368.75 | 75 | * |
| **Retrospective (Traditional Surveys)** | 192.20 | 10.00 | 700.00 | 75 | |

**Notes.**

*Indicates statistical significance at $p < .001$.

not have facilities/equipment to exercise in their homes. Moreover, 68.10% of participants reported that they have space to exercise at home, while 31.90% did not find they have space to exercise at home.

Table 2 shows the *t*-test comparing mean PA (min/day) reported *via* retrospective measures and prospective EPAs. Overall, the mean PA reported prospectively and retrospectively were 79.82 and 204.67, respectively. A paired *t*-test confirmed that this difference was statistically significant ($p < 0.001$), indicating that participants tended to report higher PA levels in retrospective self-reports compared to real-time prospective assessments. For female participants, the mean PA reported *via* prospective EPAs and retrospective measures were 74.32 and 192.20, respectively, with a significant difference ($p < 0.001$). In male participants, the mean PA reported *via* prospective EPAs and retrospective measures were 94.73 and 223.60, also indicating a significant difference ($p < 0.001$).

Table 3 presents regression models showing the factors associated with retrospectively (Model 1) and prospectively reported PA. The models include overall (Model 1a and 1b), male (Models 2a and 2b), and female (Models 3a and 3b). In the overall sample, retrospective model detected one significant association: adults who agreed to undertake PA for fun ($\beta = 0.650$, 95% confidence interval CI [0.292–1.007], $p < 0.001$) had higher PA compared to adult who disagreed. Contrastingly, the prospective model detected three significant associations: participants who agreed to undertake PA for fun ($\beta = 0.504$, CI [0.152–0.857], $p < 0.001$) had higher PA than those who disagreed. In addition, participants who disagreed with having facilities/equipment to exercise at home ($\beta = 0.514$, CI [0.127–0.900], $p < 0.001$) had higher PA than those who disagreed. Finally, citizen scientists who agreed to having space to exercise at home ($\beta = -0.400$, CI [−0.791 to −0.008], $p < 0.001$) had lower levels of PA than citizen scientists who do not have space to exercise at home. In male, the retrospective model detected one significant association: participants who agreed to undertake PA for fun ($\beta = 0.862$, CI [0.263–1.460], $p < 0.001$) had higher PA compared to participants who do not undertake PA for fun. However, no significant associations were detected in the prospective model. In the female sample, the retrospective model

detected one significant association: citizen scientists who agreed to undertake PA for fun ($\beta = 0.702$, CI [0.252–1.153], $p < 0.001$) had higher PA compared to their counterparts who do not undertake PA for fun. In the prospective model, one significant association was detected: adults who agreed to undertake PA for fun ($\beta = 0.404$, CI [0.009–0.799], $p < 0.001$) had higher PA than adults who reported disagree. The sensitivity analysis showed a reduced number of significant associations, specifically, only one significant association was detected in the model (*i.e.,* engaging in PA for fun, with no change in directionality, while other associations were not significant) (Supplemental File S1). This supports our exclusion criteria, as including participants with extreme values led to fewer detected associations.

## DISCUSSION

This study utilized a methodology that integrates ethical population health monitoring, real-time behavioral interventions, and integrated knowledge translation to engage adult citizen scientists through their personally owned smartphone devices (*Aparicio-Ugarriza et al., 2015*). It aimed to assess the difference in PA reported prospectively using EPAs and retrospectively through a modified validated survey within the same cohort. Additionally, this study investigated sociodemographic and contextual factors associated with PA reported by these two different measurement approaches. This study highlights the potential of digital citizen science as a powerful tool for health monitoring. By actively engaging participants in data collection through mobile technology, digital citizen science enhances traditional monitoring methods, enables real-time behavioral tracking, and supports scalable public health interventions (*Ibrahim, Hammami & Katapally, 2023*; *Den Broeder et al., 2018*).

The primary findings indicate a significant difference between retrospective and prospective PA reports, with retrospective measures indicating higher levels of PA. In contrast, a similar study involving a youth cohort found that prospective measures reported higher PA levels compared to retrospective measures (*Lonsdale et al., 2016*). This difference may be attributed to lifestyle variations between youth and adults, particularly regarding PA patterns. Adults generally have more structured routines, including work schedules, which may lead to well-planned PA sessions (*Aspinwall & Taylor, 1997*). Therefore, while retrospective PA reports may still lead to overestimation (*Andorko et al., 2019*; *Schaller et al., 2016*), adults might recall their PA more accurately in a structured manner. Alternatively, the spontaneous, intermittent activities common among youth might be more accurately captured through EPAs (*Zhang, Zhou & Kwan, 2019*), hence making EPAs a more appropriate method for PA monitoring in youth. Moreover, different age groups may have varying recall abilities, which can affect how PA is reported (*Joffer et al., 2019*). These differences underscore the importance of using age-appropriate PA measurement methods to ensure accurate and reliable monitoring.

In the overall sample, the retrospective PA model identified one significant association (Model 1a), whereas the prospective PA model identified three significant associations (Model 1b). In the prospective model, adults who undertook PA for fun and had access to

Hung et al. (2025), *PeerJ*, DOI 10.7717/peerj.19793

**Table 3  Regression models showing associations between retrospective and prospective measured PA and social, and health motivations of PA in the overall and gender sample.**

| | Overall | | Gender Male | | Female | |
|---|---|---|---|---|---|---|
| | Model 1: retrospective PA | Model 2: prospective PA | Model 1: retrospective PA | Model 2: prospective PA | Model 1: retrospective PA | Model 2: prospective PA |
| **Independent Variables** | | | | | | |
| Undertake PA because it's fun–Disagree | **Ref** | **Ref** | **Ref** | **Ref** | **Ref** | **Ref** |
| Undertake PA because it's fun–Agree | 0.650[***] (0.292, 1.007) | 0.504[***] (0.152, 0.857) | 0.862[***] (0.263, 1.460) | 0.692[*] (−0.103, 1.486) | 0.702[***] (0.252, 1.153) | 0.404[***] (0.009, 0.799) |
| Has facilities/equipment to exercise at home–Disagree | **Ref** | **Ref** | **Ref** | **Ref** | **Ref** | **Ref** |
| Has facilities/equipment to exercise at home–Agree | 0.108 (−0.285, 0.500) | 0.514[***] (0.127, 0.900) | −0.014 (−0.645, 0.618) | 0.667 (−0.172, 1.506) | 0.340 (−0.172, 0.851) | 0.423[*] (−0.026, 0.872) |
| To improve cardiovascular fitness–Disagree | **Ref** | **Ref** | **Ref** | **Ref** | **Ref** | **Ref** |
| To improve cardiovascular fitness–Agree | 0.047 (−0.462, 0.557) | 0.336 (−0.166, 0.838) | 0.684[*] (−0.094, 1.462) | 0.384 (−0.648, 1.417) | −0.424 (−1.110, 0.262) | 0.489 (−0.113, 1.091) |
| Space to exercise at home–Disagree | **Ref** | **Ref** | **Ref** | **Ref** | **Ref** | **Ref** |
| Space to exercise at home–Agree | 0.218 (−0.179, 0.615) | −0.400[***] (−0.791, −0.008) | −0.019 (−0.675, 0.637) | −0.907[*] (−1.778, −0.035) | 0.078 (−0.440, 0.597) | −0.222 (−0.677, 0.233) |
| Maintain physical health–Disagree | **Ref** | **Ref** | **Ref** | **Ref** | **Ref** | **Ref** |
| Maintain physical health–Agree | 0.661 (−0.145, 1.468) | 0.352 (−0.444, 1.147) | 1.683[***] (0.168, 3.199) | 1.533 (−0.480, 3.545) | 0.460 (−0.506, 1.426) | −0.041 (−0.888, 0.806) |
| Constant | 4.158[***] (3.274, 5.041) | 3.215[***] (2.343, 4.086) | 2.852[***] (1.279, 4.426) | 2.261[***] (0.172, 4.349) | 4.648[***] (3.571, 5.725) | 3.412[***] (2.467, 4.357) |
| Observations[β] | 105 | 105 | 36 | 36 | 69 | 69 |

**Notes.**

[***] Significant at $p < 0.05$; All models controlled for age, gender, education, and employment status.

[β] Some observations were deleted due to missingness.

[*] Indicates statistical significance at $p < .001$.

facilities or equipment reported higher PA levels, while those with space to exercise at home reported lower PA levels. Conversely, in the retrospective model, adults who engaged in PA for fun reported higher levels of PA compared to those who did not. These findings may suggest that intrinsic motivation for PA plays an important role in adult engagement in PA. Previous research has shown that intrinsic motivation is a strong predictor of sustained PA (*Esmaeilzadeh, Rodriquez-Negro & Pesola, 2022*; *Brunet & Sabiston, 2011*; *Dacey, Baltzell & Zaichkowsky, 2008*). For instance, a study found that adults who maintained regular PA for at least 10 years reported higher intrinsic motives compared to those who remained sedentary or reduced their PA levels (*Geller et al., 2018*). Similarly, another study noted a positive correlation between intrinsic motivation and PA across various adult age groups (*Molanorouzi, Khoo & Morris, 2015*). These consistent findings highlight the robust influence of intrinsic motivation on PA behavior. Furthermore, an important consideration to take into account is that the concept of fun can be inherently subjective and may be interpreted differently based on personal preferences, cultural background, or past experiences with PA. This subjectivity could influence how respondents report their motives for engaging in PA, which introduces variability in the association between enjoyment and PA levels (*Creighton et al., 2022*). Thus, future studies could benefit from incorporating qualitative approaches to better capture the diverse ways adults conceptualize fun in the context of PA.

Moreover, participants who reported having adequate facilities and equipment for exercise at home showed higher levels of PA. This finding is in line with other similar studies, which state that access to exercise facilities is associated with increased levels of moderate-to-vigorous PA (*Mok et al., 2019*). Accessibility to exercise resources has been reported to play a critical role in fostering an active lifestyle and improving PA levels among adults (*DeVries, Baselmans & Bartels, 2021*). Having access to exercise facilities at home could facilitate higher PA accumulation (*Perski et al., 2022*), especially among adults who experience time constraints due to their busy work schedules (*Park et al., 2022*). Additionally, the presence of PA facilities or equipment at home removes travel time to and from the gym, which has been reported as a main barrier for adult participation in PA (*Mok et al., 2019*). These findings emphasize the significance of the physical environment in encouraging PA participation and suggest that focusing on the availability of home exercise facilities can help overcome barriers to PA.

Having access to space for exercise at home was associated with lower levels of PA compared to those without space. This contrasts with findings from other studies, which have shown that having more space at home is associated with higher levels of PA, as it provides a convenient and accessible environment for exercise, reducing barriers such as travel time and facilitating more frequent and consistent PA (*Meghani et al., 2023*). One possible explanation for these findings could be a loss of intrinsic motivation to engage in PA, as individuals may be less inclined to use their home exercise space effectively without external encouragement (*Dishman et al., 2018*). Likewise, research has shown a positive relationship between autonomous motivation (*i.e.,* motivation driven by personal interest, enjoyment, and values) and exercise (*Teixeira et al., 2012*). Additionally, the lack of social support and accountability, which can be crucial

for sustained PA, may lead to lower PA frequency (*Deci & Ryan, 2000*), even when adults have access to space for exercise. Exercising at home can induce feelings of isolation among individuals (*Mays et al., 2021*), potentially reducing motivation, adherence to a PA routine, and consistency. Social support has been shown to be a key driver of PA engagement, and individuals lacking external reinforcement from peers, workout partners, or structured group settings may be less likely to engage in regular exercise (*Meghani et al., 2023*; *Lieber et al., 2024*; *Smith, Moyle & Burton, 2023*). Furthermore, the home environment often contains numerous distractions, including ringing doorbells and children making noise (*Toniolo-Barrios & Pitt, 2021*; *Moretti et al., 2020*), which can increase screen time and reduce opportunities for PA (*Tandon et al., 2014*). The ease of access to entertainment and relaxation activities (*i.e.,* television viewing) at home can make it challenging to prioritize PA (*Fingerman et al., 2021*). These findings suggest that, while home exercise spaces offer convenience, they may inadvertently reduce PA levels due to a lack of social reinforcement and the presence of distractions. Creating structured, dedicated, and socially engaging spaces could be key to minimizing distractions in the home environment and promoting sustained exercise behavior in adults. Future research should further investigate this finding using a larger sample size.

In the gender-segregated model, female participants who reported undertaking PA for fun had higher PA levels than those who did not, both retrospectively and prospectively. This underscores the influence of intrinsic motivation on PA behaviors among females. Studies have shown that females often react to contextual demands (*i.e.,* social and environmental cues or expectations, such as cultural norms or situational challenges) in ways that emphasize intrinsic motivators (*Khalid, 2024*; *Segar, Eccles & Richardson, 2008*), and they report higher levels of intrinsic motivation that encourage engagement in PA (*Cabras et al., 2023*). Intrinsic motivation plays a critical role in fostering long-term behavioral change (*Teixeira et al., 2012*), which can include integrating a consistent PA routine into one's schedule. For instance, females, engaging in activities based on personal interest, were more likely to adhere to and maintain long-term weight loss programs (*Williams et al., 1996*). Yet, some studies suggest that extrinsic motivation may have a stronger influence on PA than intrinsic motivation in females (*Segar, Spruijt-Metz & Nolen-Hoeksema, 2024*). These findings highlight the complexity of motivational factors in PA behavior among females, emphasizing the need to create environments that make PA enjoyable and valuable to an individual, to support sustained PA and overall well-being. The stronger association of intrinsic motivation with PA in females may reflect societal and cultural factors, such as the emphasis on personal fulfillment and relational well-being, which are often more strongly encouraged in females (*O'Dougherty, Kurzer & Schmitz, 2010*; *Alajlan, Aljohani & Boughattas, 2024*). Additionally, psychological differences in how females perceive and respond to motivational cues, such as a greater focus on internal satisfaction, could further explain these findings (*O'Dougherty, Kurzer & Schmitz, 2010*). Supplemental File S2A depicts the key relationships between PA and various motivational and contextual factors in females.

Conversely, males who engaged in PA for fun and to maintain physical health had higher PA levels than those who did not. With physical health being an external goal

(*i.e.,* goals driven by external rewards or pressures, such as social recognition or fear of negative outcomes) pursued by the means of PA, engaging in PA to maintain physical health reflects extrinsic motivation. This finding indicates the significance of both intrinsic and extrinsic motivation in driving PA behaviors among males. Moreover, research has found that males are often motivated by intrinsic factors such as self-determined motivation, which significantly contribute to their PA engagement (*Brunet & Sabiston, 2009*). Simultaneously, males also exhibit higher levels of extrinsic motivation, particularly through competitiveness (*Kilpatrick, Hebert & Bartholomew, 2005*). For instance, studies have shown that males score higher on competitiveness and focus on external outcomes, such as outperforming others (*Gill & Dzewaltowski, 1988*). These motivational factors are crucial for understanding PA behaviors in males and suggest that both personal satisfaction and external rewards are important for sustaining PA. The stronger influence of extrinsic motivators in males may be tied to societal norms that encourage males to prioritize physical health, strength, and competition, as well as psychological tendencies to seek external validation or achievement (*Molanorouzi, Khoo & Morris, 2015*; *Liu et al., 2023*). These gender-specific motivational patterns underscore the importance of tailoring PA interventions to address the unique drivers of behavior in males and females. Appendix S2b depicts the key relationships between PA and various motivational and contextual factors in males.

The sensitivity analysis, which included outliers, revealed a reduction in the number of significant associations compared to the primary regression model. Specifically, while the original model identified significant associations between PA participation and factors such as engaging in PA for enjoyment, having access to exercise facilities or equipment at home, and having space to exercise at home, the sensitivity analysis retained significance only for engaging in PA for enjoyment. This reduction in significant associations suggests that certain data points exert disproportionate influence on the regression estimates, highlighting the role of influential outliers in skewing parameter estimates and inflating standard errors, potentially leading to biased estimates (*Sullivan, Warkentin & Wallace, 2021*; *Meghani, Byun & Chittams, 2014*). Including outliers can result in inaccurate maximum likelihood estimates, thereby compromising the validity of statistical inferences (*Zanin, Lóczi & Zanin, 2024*). In our analysis, the persistence of the association between engaging in PA for enjoyment, even after accounting for outliers, indicates that this relationship is robust and less susceptible to the influence of extreme data points. In contrast, the loss of significance in other associations suggests that they may be more sensitive to data variability or potential measurement errors. Consequently, cautious interpretation is warranted for associations that are not robust to the inclusion of outliers, and future research should consider employing robust regression techniques or conducting further sensitivity analyses to assess the stability of these associations, as recommended in contemporary statistical practice.

Our findings not only focus on the difference between retrospective and prospective measures of PA, but it also shed light on the nuanced associations between sociodemographic and contextual factors with PA accumulation in adults. These insights call for a critical reevaluation of current public health strategies to promote PA. For instance,

the gender-specific motivational drivers identified in this study suggest that a one-size-fits-all approach to PA interventions is unlikely to be effective. Policymakers should prioritize funding and developing gender-tailored programs. For females, initiatives could focus on creating community-based PA opportunities that emphasize enjoyment, social connection, and intrinsic rewards (*e.g.*, group fitness classes or recreational leagues), while for males, programs could incorporate goal-oriented challenges, competitive elements, and health outcome tracking to align with their extrinsic motivators. Additionally, the discrepancy between retrospective and prospective PA measures underscores the need for policymakers to invest in prospective tools to capture real-time PA behaviors and contextual factors. This data can inform the design of adaptive interventions that respond to individuals' changing needs and environments. Policymakers should also mandate the inclusion of prospective PA measures in national health surveys and public health evaluations to ensure a more accurate understanding of PA patterns and trends. As digital technologies continue to evolve, future research should explore these dynamics further and strive to refine PA measurement tools. Tailoring interventions to different lifestyles and demographics will be essential. In this digital age, the potential of EPAs and other digital technologies is increasingly apparent, offering promising avenues for personalized and effective strategies to promote PA across various populations.

However, for these digital approaches to have a global impact, they must be adapted for use in low-resource settings. In the context of digital citizen science, this involves developing tools that are accessible to communities with limited internet connectivity and lower digital literacy. Strategies such as simplified mobile applications, offline data collection capabilities, and partnerships with community organizations could help bridge the digital divide and ensure accessibility for populations with limited technological infrastructure. These approaches are particularly important for ensuring the inclusion of digitally underserved populations, who may otherwise face barriers to participating in digital citizen science initiatives. Enhancing digital literacy is essential not only for engaging with research tools but also for accessing critical resources and services, as individuals with lower digital proficiency may struggle to navigate digital platforms effectively, limiting their ability to utilize health-promoting technologies (*Hung & Katapally, 2025*). Implementing offline data collection capabilities is crucial in low-resource settings, where limited internet connectivity can hinder real-time tracking of PA behaviors, making it necessary to ensure continuous and accurate data collection across diverse environments (*Reichold et al., 2021*). Given that the participants in this study were drawn from a single geographic area (Regina, Saskatchewan), the findings may not fully translate to other populations or settings with different socio-economic, technological, or cultural contexts. To enhance the generalizability of digital citizen science approaches, future research should explore how these methods can be adapted to diverse environments, particularly low-resource settings where technological infrastructure and digital literacy may differ significantly.

Digital citizen science tools should be explored in various contexts, examining factors such as demographic differences, engagement patterns, and technological accessibility. Investigating these aspects would provide a more comprehensive understanding of how digital PA tracking can be optimized for broader public health applications. Additionally,

as digital health technologies continue to evolve, ensuring ethical data collection remains a critical consideration (*Garett & Young, 2022*). Recent studies on ethical health monitoring emphasize the importance of secure data management, participant autonomy, and transparent data-sharing practices, particularly in mitigating risks such as privacy breaches, exclusion of certain groups from monitoring, and potential stigmatization from data sharing (*Klingler et al., 2017*). Dynamic consent mechanisms, which allow participants to modify their consent preferences in real-time, have been proposed as a means to enhance user control and trust in digital health research (*Prinsen, 2024*). Moreover, concerns around authentication security, data sovereignty, and transparent communication highlight the need for user-centered privacy protections and trust-building measures in digital health platforms (*Lee et al., 2024*). While the Smart Platform incorporates robust encryption, participant data ownership, and real-time consent review, future research should explore ways to further strengthen privacy safeguards, enhance user control over data-sharing, and develop more accessible security measures to ensure inclusivity across diverse populations.

## Strengths and limitations

This study presents several strengths, including the innovative use of digital citizen science to advance our comprehension of digital tools and methodologies for health monitoring. By leveraging participant-owned smartphones for ethical and real-time PA tracking, this study demonstrates how digital citizen science can enhance PA measurement accuracy, reduce recall bias, and engage individuals directly in health research. The findings of this study contribute to new evidence regarding the digitalization of measures, both traditionally and prospectively, at participants' convenience. Furthermore, this study is one of the few studies comparing traditional surveys with prospective EPAs among an adult cohort. The limitations of this study include recall bias, where participants may not have accurately reported PA levels due to difficulties in remembering past activities or unintentional misestimation. In this study, prospective and retrospective PA assessments were examined during an eight-day period. However, this duration may not sufficiently capture the variability in PA levels. Previous research has shown distinct activity patterns between weekdays and weekends, with some individuals being more active on specific days, while others maintain consistent or low activity levels (*Suorsa et al., 2023*). Additionally, seasonal variations significantly impact PA, with higher activity levels in summer and lower levels in winter, influencing overall trends (*Garriga et al., 2021*). Future research should extend the study period to better account for fluctuations in PA and provide stronger longitudinal insights.

Another limitation is that participants were primarily from Regina, Saskatchewan, which may impact generalizability, as differences in urbanization, cultural diversity, and access to digital tools in other regions could influence PA reporting and engagement with digital citizen science. Moreover, recruitment was limited to individuals with access to digital devices, potentially excluding those with lower digital literacy or limited smartphone access. This introduces selection bias, which occurs when the study sample is not representative of the broader population (*Hegedus & Moody, 2010*), as individuals who are more comfortable with technology may engage differently with mobile devices than those with limited digital

proficiency. Future research should explore the applicability of this tool across diverse socioeconomic and cultural contexts to assess its adaptability and effectiveness in varied populations. Expanding the study to include individuals with lower digital literacy and different technological access levels would provide a more comprehensive understanding of how digital citizen science can support equitable and inclusive health monitoring. Furthermore, we adapted our PA measures from the IPAQ for deployment *via* smartphones. However, we acknowledge that these adapted formats have not been formally validated *via* smartphones. Future research should aim to compare the measurement properties of smartphone and non-smartphone- deployed IPAQ to assess PA behaviors.

## CONCLUSION

Physical inactivity is a global health concern and plays a critical role in the contribution to NCDs. Understanding patterns of PA through rigorous and validated tools is crucial in developing appropriate policies and interventions that can reduce the negative health impacts of sedentary lifestyles. The findings of this study do not discount retrospective measures, rather it shows the importance of using prospective EPAs to capture PA more accurately and in real time. Digital citizen science offers a scalable and participatory approach to health monitoring by enabling individuals to contribute real-time data through their own digital devices. With the widespread adoption of smartphones, these ubiquitous tools can be leveraged to capture accurate active living patterns of large populations through innovative EPAs. By integrating citizen-driven data into monitoring systems, public health agencies can monitor PA trends in real time, inform population-level health campaigns, and refine policies that promote active living. Furthermore, digital citizen science facilitates community engagement, personalized feedback, and behavioral interventions, making it a powerful tool for public health promotion. Ensuring ethical implementation, robust data security, and equitable accessibility will be essential to maximizing the potential of digital citizen science in shaping future public health strategies. To integrate digital citizen science into ongoing public health surveillance, these platforms should be designed to easily connect with existing health data systems, use standardized formats that make data easy to share and compare, and be supported by public health institutions to ensure long-term use and impact. Future research should also explore how EPAs can be adapted for use in global or rural settings, such as through offline capabilities, multilingual support, and simplified app interfaces, and how longitudinal designs can capture seasonal variation, life-stage transitions, and long-term behavior change to advance understanding of PA patterns over time.

### Funding

This study was funded by the Canada Research Chair which supports Tarun Katapally's work. The funders had no role in study design, data collection and analysis, decision to publish, or preparation of the manuscript.

### Grant Disclosures

The following grant information was disclosed by the authors:
Canada Research Chair.

### Competing Interests

The authors declare there are no competing interests.

### Author Contributions

- Caitlin Hung performed the experiments, analyzed the data, prepared figures and/or tables, authored or reviewed drafts of the article, and approved the final draft.
- Sheriff Tolulope Ibrahim performed the experiments, analyzed the data, authored or reviewed drafts of the article, and approved the final draft.
- Tarun Reddy Katapally conceived and designed the experiments, authored or reviewed drafts of the article, and approved the final draft.

### Human Ethics

The following information was supplied relating to ethical approvals (i.e., approving body and any reference numbers):

Research Ethics Boards of Universities of Saskatchewan and Regina (REB #2017-029).

### Data Availability

The anonymized dataset is available at figshare: Hung, Caitlin; Ibrahim, Sheriff; Katapally, Tarun (2024). Exploring ethical surveillance of physical activity behaviors: a smart platform study operationalizing digital citizen science. figshare. Dataset. https://doi.org/10.6084/m9.figshare.27038293.v3.

### Supplemental Information

Supplemental information for this article can be found online at http://dx.doi.org/10.7717/peerj.19793#supplemental-information.

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
