# Peer review of "Exploring ethical monitoring of physical activity behaviors among adults: a Smart Platform study operationalizing digital citizen science"

_PeerJ, doi:10.7717/peerj.19793_

## Round 0.1 · original submission · Major Revisions

Please address all the comments of the 3 reviewers

·

Basic reporting

no comment

Experimental design

no comment

Validity of the findings

no comment

Additional comments

The manuscript titled "Exploring Ethical Surveillance of Physical Activity Behaviors Among Adults: A Smart Platform Study Operationalizing Digital Citizen Science" investigates the potential of ethical, citizen-driven digital tools to measure physical activity (PA). The study bridges a critical gap between traditional retrospective PA measures and ecological prospective assessments (EPAs) using smartphones. While the work is novel and methodologically sound, it requires substantial revisions to enhance clarity, address methodological gaps, and improve its overall impact. Below are detailed suggestions for improvement.
1 The manuscript could better situate its research within existing literature. For example, while it mentions gaps in comparing retrospective and prospective measures, more in-depth discussions of prior works on EPAs and their application across diverse populations would strengthen the rationale. And add references to studies involving similar methodologies to support the novelty claim.
2 The study's participants primarily represent a single geographic region (Regina, Saskatchewan). It would be beneficial to discuss how this might limit the generalizability of findings and address potential biases in participant recruitment.
3 While the regression analyses provide valuable insights, the justification for excluding participants with extreme PA values could be elaborated. Clarify how this exclusion impacts the overall analysis and whether sensitivity analyses were performed.
4 The results for gender-segregated models need further interpretation, especially regarding why some associations are significant for one gender but not the other.
5. The ethical framework supporting the use of citizen-owned devices for PA tracking is well-articulated. However, potential privacy risks and mitigation strategies (e.g., encryption methods, data deletion requests) should be more comprehensively detailed, as ethical surveillance is a central theme.
6. The manuscript highlights significant differences between retrospective and prospective PA measures. However, it does not delve deeply into the practical implications of these findings for public health policy or intervention design. Strengthen the discussion by suggesting actionable recommendations.
7. Improve the clarity of Figures 1 and 2 by including additional labels or annotations to guide interpretation. Ensure consistency in formatting and enhance the visual appeal of Table 3 to make it easier to interpret regression results.
8. Provide additional details about the development and validation of the smartphone-based EPA tool. Readers might question its reliability and whether it sufficiently captures diverse PA behaviors.
9.Clarify the time-triggering mechanism used for EPA notifications and whether it influenced participant compliance.
10.The conclusion would benefit from a stronger emphasis on how this research can guide the integration of digital citizen science into broader public health initiatives.
11. Include specific recommendations for future studies, such as testing the tool across diverse socioeconomic or cultural contexts or extending the study duration for longitudinal insights.

Reviewer 2 ·

Basic reporting

Language and Clarity: The manuscript is written in professional and clear English, with minimal grammatical errors. The text is generally accessible to an academic audience, though some technical sections could benefit from simplification to enhance readability for non-specialists.

Literature References: The manuscript provides sufficient references to foundational and recent literature on physical activity measurement and digital citizen science. However, including more recent studies on ethical data collection in health surveillance would strengthen the context.

Structure and Presentation: The article follows a logical structure with well-organized sections. Figures and tables are relevant and of high quality but could be better annotated for clarity.
The raw data is shared, which is commendable and aligns with transparency requirements. However, clearer documentation of the dataset's variables would enhance usability for future replication.

Self-Contained: The manuscript is self-contained, with results directly relevant to the research questions. The integration of findings with hypotheses is clear and consistent.

Experimental design

Scope and Research Gap: The study is well within the journal's scope, addressing an important and timely topic in public health. The research question is clearly defined, focusing on the comparison of retrospective and prospective physical activity measures and their contextual correlates.

Methodological Rigor: The study is conducted to a high ethical standard, with robust privacy measures and participant consent processes. The use of a digital citizen science platform is innovative and aligns with the stated objectives.
The recruitment and inclusion criteria are well-described, but the rationale for the seven-day study duration could be expanded. Explain whether this period sufficiently captures variability in physical activity patterns.

Replicability: The methodology is described in sufficient detail to allow replication. However, the dichotomization of some variables (e.g., motivational and environmental factors) may obscure nuances in the data. Justify this decision or consider providing additional analyses with continuous variables.

Validity of the findings

Data Robustness:
The data analysis is statistically sound and clearly reported. The comparison of retrospective and prospective measures is well-supported by paired t-tests, and the regression models adequately address the research questions.
The treatment of outliers and missing data is appropriate, but the handling of potential biases (e.g., social desirability in self-reports) could be discussed further.

Conclusions:
The conclusions are well-aligned with the findings and research questions. The authors highlight the utility of prospective measures for more accurate physical activity surveillance, supported by statistically significant differences and associations.
Recommendations for future research are relevant but could be expanded. For instance, the authors might propose longer study durations or explore differences across diverse demographic groups.

Additional comments

The manuscript introduces the innovative use of digital citizen science in health surveillance, a strength that should be emphasized more in the discussion.
The ethical considerations are well-addressed, but a brief discussion on how these methods could be adapted for use in low-resource settings would enhance the global relevance of the study.
The finding that participants with home exercise spaces report lower physical activity warrants further exploration. Potential behavioral or environmental mechanisms should be discussed more thoroughly.

Annotated reviews are not available for download in order to protect the identity of reviewers who chose to remain anonymous.

·

Basic reporting

First of all, congratulations for your research. It is a comprehensive and good research, but I have some suggestions to improve the article. The introduction of the article provides a comprehensive framework on the importance of physical activity and measurement methods, but I think there are missing points.

Although the introduction draws attention to the problems of current measurement methods and the gaps in the measurement of physical activity, it is not clear how this research will fill these gaps. The innovative aspects of this research could be emphasised more clearly. In addition, it may be useful to mention issues such as physical activity states and physical activity barriers as a small paragraph. I'm attaching a few articles as examples, but you don't have to use them. I suggest them to guide you.
https://e-jespar.com/index.php/jespar/article/view/7/4
https://e-jespar.com/index.php/jespar/article/view/2/8
https://e-jespar.com/index.php/jespar/article/view/6/5

Digital citizen science and ethical oversight are emphasised, but how these approaches are superior to existing methods and what specific advantages they provide should be detailed.
When I examined the study, although the literature review seems to have been done comprehensively, a more in-depth coverage of the limitations of existing studies with a critical perspective may make the contribution of this study more evident.
The aim of the study in the introductory part of the study

Experimental design

The experimental design was well planned but was the sample size determined by power analysis?

Validity of the findings

The findings and analyses seem unproblematic.

---

## Round 0.2 · Minor Revisions

Please respond to the reviewer's minor comments.

Reviewer 2 ·

Basic reporting

The manuscript is clearly written, professionally presented, and conforms to PeerJ’s formatting expectations. The background is thorough and well contextualized, drawing from relevant and recent literature on digital citizen science, physical activity (PA) surveillance, and ecological prospective assessments (EPAs). The figures are well-labeled and appropriately referenced, and the tables provide clear summaries of results.

The authors have supplied a detailed account of the ethical approval process, and the raw data are reported to be available, although it would help if these were better referenced in the main text (e.g., as supplemental material). The language is fluent and accessible to an international academic audience, and the structure is logical and easy to follow.

Minor suggestion: The introduction could be shortened slightly by merging overlapping content (especially lines 134–148 and 150–164) for improved readability.

Meets expectations.

Experimental design

The study is within the journal’s scope and offers original primary research. The research questions are well defined, timely, and address a real-world issue: the need for more valid and ethical methods of measuring physical activity using ubiquitous digital tools.

The design—comparing retrospective and prospective PA reporting via smartphones—is innovative and relevant. The study clearly explains how it fills a knowledge gap, and the Smart Platform is described with sufficient technical detail for replication. The use of the modified IPAQ is appropriate, although the authors do acknowledge that the adapted smartphone version has not been formally validated, which is noted as a limitation.

Recruitment, inclusion/exclusion criteria, and bias mitigation strategies are clearly explained. Ethical procedures, consent, and participant data rights are addressed with commendable detail.

Well described and ethically conducted.

Validity of the findings

The conclusions are well grounded in the data. The statistical analysis is robust and appropriately performed using open-source software (R). Given the research questions, paired t-tests and linear regression models are suitable.

Key findings are:

Retrospective reports of PA are consistently higher than prospective ones.

Motivation (especially fun) is significantly associated with PA in both models.

Gender differences in motivation (intrinsic vs. extrinsic) are thoughtfully explored.

The discussion is reflective and situates the findings within broader literature. Limitations—such as short measurement period, localized sample, and lack of formal validation for the smartphone-deployed IPAQ—are transparently acknowledged. The authors offer realistic and well-informed suggestions for future research and practical applications.

The conclusions are statistically sound supported.

Additional comments

This is a well-designed and thoughtfully executed study that contributes valuable insights into digital citizen science and physical activity (PA) monitoring. The use of both retrospective and prospective self-reports via smartphones is innovative and timely, especially considering the increasing reliance on mobile health tools for public health research and intervention. The integration of ethical considerations, real-time data collection, and participant engagement is particularly commendable.

The manuscript is generally well-written, methodologically sound, and grounded in relevant literature. However, several areas could be strengthened to further improve clarity, transparency, and overall impact:

1. Clarify the adaptation of the IPAQ instrument:
While the IPAQ is a widely accepted tool, more detail is needed about how it
was modified for deployment via smartphone. Please clarify:

- What specific changes were made to question wording, structure, or response options?

- Was this adapted version pre-tested or validated?

- How might these changes have affected response reliability or comparability with other IPAQ-based studies?

2. Explain the sensitivity analysis more fully:
The sensitivity analysis (Appendix 1) is a useful inclusion, but its interpretation could be clearer. Consider expanding on:

- Why were fewer significant associations found when including outliers?

- What does this suggest about data variability or reliability?

- How should readers interpret the robustness of your main findings in light of this?

3. Clarify the term “ethical surveillance”:
This is a compelling concept, but the term could be misunderstood. Please provide a clearer definition earlier in the manuscript:

- How is “ethical surveillance” distinct from conventional surveillance?

- What are the specific ethical safeguards that support this framing (e.g., participant consent, data ownership, ability to withdraw)?

4. Expand on participant engagement as citizen scientists:
The manuscript briefly notes two-way communication between participants and researchers. Could you elaborate on:

- How frequently this occurred, and through what means (e.g., app messaging, email)?

- Whether this feedback was formally documented and influenced ongoing procedures?

- How this participatory process enhanced the validity or ethics of the study?

5. Address digital literacy and accessibility concerns:
As the study relies on participants using smartphones, it would be useful to reflect on:

- Whether digital literacy or familiarity influenced participation or data quality.

- What onboarding or support was provided to participants unfamiliar with the app.

- How future studies could ensure inclusion of digitally underserved populations.

6. Discuss generalizability beyond the current sample:
Given that participants were drawn from one geographic area (Regina, Saskatchewan), please consider addressing:

- To what extent these findings might translate to other populations or settings.

- How digital citizen science might be adapted for low-resource environments or areas with limited internet/smartphone access.

7. Clarify how “fun” as a motivational factor was measured:
The finding that “undertaking PA for fun” is associated with higher PA levels is central to your argument. Please explain:

- Was this measured via a single item or part of a larger validated scale?

- How might subjectivity or individual interpretation of “fun” influence responses?

8. Add a visual summary or conceptual model:
A figure or logic model summarizing key relationships (e.g., motivational and contextual factors → PA behaviors, stratified by gender) would aid comprehension and reader engagement. This could be placed in the discussion or conclusion to reinforce the implications of the findings.

9. Improve visibility of raw data availability:
While you note the use of open data and open-source software, make sure readers know:

- Where the dataset and scripts are hosted (e.g., supplemental material, external repository).

- What formats and metadata are available for reuse and replication.

10. Condense overlapping sections in the introduction:
The background is thorough but occasionally repetitive. For instance, the rationale between lines 134–148 and 150–164 could be consolidated to streamline your argument and improve clarity.

11. Add future research directions:
Your conclusion could be strengthened by suggesting more specific next steps. For example:

- What adaptations would make EPAs more accessible in global or rural settings?

- How can digital citizen science be integrated into ongoing public health surveillance efforts?

- What types of longitudinal designs would build on your findings?

Overall, this is a strong manuscript with excellent potential for impact. The above suggestions are offered to help sharpen the presentation, clarify methods, and deepen engagement with key concepts and future implications. Thank you for this valuable contribution to the literature.

---

## Round 0.3 · accepted · Accept

Thank you for your revised submission. I am satisfied that you have addressed the remaining concerns of the reviewers, and am happy to accept your paper for publication.